# Research on the Primary Frequency Regulation Control Strategy of a Wind Storage Hydrogen-Generating Power Station

**Dongyang Sun** [1]**, Wenyuan Zheng** [1,*]**, Jixuan Yu** [1] **and Ji Li** [2]

1   School of Electrical and Electronic Engineering, Harbin University of Science and Technology, Harbin 150080, China
2   Electric Power Research Institute of State Grid Xinjiang Electric Power Co., Ltd., Urumqi 830011, China
*   Correspondence: 2120300068@stu.hrbust.edu.cn

**Abstract:** Wind curtailment and weak inertia characteristics are two factors that shackle the permeability of wind power. An electric hydrogen production device consumes electricity to produce hydrogen under normal working conditions to solve the problem of abandoning wind. When participating in frequency regulation, it serves as a load reduction method to assist the system to rebuild a power balance and improve the wind power permeability. However, due to its own working characteristics, an electric hydrogen production device cannot undertake the high-frequency component of the frequency regulation power command; therefore, an energy storage device was selected to undertake a high-frequency power command to assist the electric hydrogen production device to complete the system frequency regulation. This paper first proposes and analyzes the architecture of a wind storage hydrogen-generating station for centralized hydrogen production with a distributed energy storage, and proposes the virtual inertia and droop characteristic mechanism of the wind storage hydrogen-generating station to simulate a synchronous unit. Secondly, an alkaline electrolysis cell suitable for large-scale engineering applications is selected as the research object and its mathematical model is established, the matching between different energy storage devices and their cooperation in power grid frequency regulation is analyzed, and a super capacitor is selected. A control strategy for the wind storage hydrogen-generating power station to participate in power grid frequency regulation with a wide time scale is then proposed. Using the first-order low-pass filter, the low-frequency component of the frequency regulation power command is realized by an electric hydrogen production device load reduction, and a high-frequency component is realized by the energy storage device. Finally, the effectiveness and rationality of the proposed control strategy are verified by establishing the simulation model of the wind storage hydrogen-generating power station with different initial wind speed states, comparing the system frequency dip values under the proposed multi-energy cooperative control strategy and a single energy device control strategy.

**Keywords:** doubly fed induction generator (DFIG); power-to-hydrogen (P2H); super capacitor energy storage system (SCESS); primary frequency regulation; multi energy cooperation

## 1. Introduction

In 2021, a Chinese government work report clearly pointed out that it will strive to achieve a "carbon peak" by 2030 and "carbon neutrality" by 2060 [1]. By then, the total installed capacity of wind power and solar power is expected to exceed 1.2 billion kilowatts [2]. With the increasing penetration of wind power, the shortcomings of wind turbines in grid-connected operations are becoming more and more obvious [3]. Different from conventional synchronous units with a frequency regulation capability [4], wind turbines operate according to the maximum wind power curve, and the rotor speed is completely decoupled from the system frequency, so that it does not have the ability to respond to system frequency changes [5]. This shortcoming of a weak frequency regulation ability aggravates the source-load unbalance of the system, and may lead to the collapse

of the power system in severe cases, which brings great challenges to the safe and stable operation of a power system [6]. Therefore, in order to solve the system stability problem caused by a large-scale grid connection of wind power, the power grid guidelines issued in China clearly require that wind farms should have the same primary frequency regulation capability as conventional power stations [7].

Domestic and foreign scholars have carried out significant research on the problem of wind power frequency regulation, mainly by actively releasing rotor kinetic energy, abandoning an MPPT state to reserve backup power, and adding energy storage devices to give wind turbines the ability to respond to system frequency changes [8,9]. References [10,11] proposed a primary frequency regulation method for simulating synchronous units, which used an integrated inertial control to change the power output of wind turbines; however, this control method would change the original operating state of a wind turbine, which is not conducive to the stable operation of a system. References [12,13] sacrificed the maximum wind energy tracking state, and a rotor speed and pitch angle control strategy was used to provide a system with standby frequency regulation power, but the wind energy utilization rate was reduced. References [14,15] proposed a coordinated control strategy, but the operating conditions of each doubly fed induction generator (DFIG) were different, it was difficult to realize the coordinated control of multiple units and the control system was cumbersome.

Meanwhile, meeting the energy demand of a load has been proposed in the literature, through an energy release by attaching an energy storage device [16]. Reference [17] proposed an architecture where the energy storage device was connected in parallel to both ends of a DC bus of a DFIG, while references [18,19] proposed adding energy storage devices on the basis of wind turbines to jointly participate in a system's frequency regulation, but a single energy storage device could not meet the full time scale frequency regulation requirements. Additionally, reference [20] proposed a joint inertial support method based on the additional control of a wind turbine as well as additional energy storage devices, but the required configuration capacity of the energy storage devices was too large considering the primary frequency regulation, which reduced the system's economy.

The vigorous development of power-to-hydrogen (P2H) technology provides a new idea for grid frequency regulation [21]. On the one hand, as an effective means to solve the problem of wind abandonment, a P2H device can improve the wind power absorption capacity, producing hydrogen from abandoned wind to improve the economy of a system's operation [22]. On the other hand, P2H can be used as a large-capacity controllable load, that has the advantage of a large energy conversion volume, which is of great significance for improving the frequency stability of the power grid [23]. The energy storage system quickly exchanges energy with the system when the system's power balance is broken, and it assists the system to rebuild the power balance [24]. Responding to system frequency changes through P2H devices and energy storage devices provides a new idea for the frequency regulation of wind farms [25]. Reference [26] defined the inertia of a wind power energy storage system based on the inertia characteristics of synchronous units, they calculated the energy storage capacity of an auxiliary wind farm for frequency regulation, and used the fuzzy logic control algorithm to propose a control strategy that used an energy storage device to compensate for the inertia of the wind farm; however, the sag characteristics were not considered. Reference [27] established the super capacitor energy storage system–doubly-fed induction generator (SCESS-DFIG) architecture for wind turbine-attached energy storage devices, and proposed a power fluctuation suppression method combining a wind turbine itself with energy storage devices, but the required capacity configuration of the SCESS was too high, which would reduce the economic efficiency of wind farms. References [28,29] regarded P2H as an effective means to solve the problem of wind curtailment, and regarded it as a controllable load to participate in the frequency adjustment of a system, but they did not consider the frequency regulation capability of an electric hydrogen production device itself. Reference [30] proposed a power compensation scheme that would adjust the frequency of a participating system. The P2H and SCESS would work

together to ensure that the grid-connected power was consistent with the load dispatching, but this does not explain the relationship between a grid frequency change and the required power compensation. Reference [31] proposed a microgrid architecture based on the joint management of the supply side of new energy generation as well as the load side, but the control and management of energy for complex energy architectures is a difficult research point [32]. Reference [33] proposed a rotating coordinated control strategy of an electrolytic cell array to meet the absorption capacity and effectively improve the service life of the electrolytic cell array, while they also completed a simulation verification based on the actual case of a wind farm in Zhangbei County, but its off-grid architecture could not reflect the role of an electric hydrogen production device as a controllable load reduction device to participate in frequency regulation.

In view of the above problems, this paper proposes a super capacitor energy storage system–doubly-fed induction generator–power-to-hydrogen (SCESS-DFIG-P2H) generating station architecture with centralized hydrogen production and distributed energy storage. Consequently, a cooperative frequency regulation strategy based on a P2H unit and a SCESS unit is proposed.

(1) Under normal conditions, as an important means to solve the problem of wind curtailment, the P2H device consumes the electric energy outside the grid-connection of the DFIG, and the hydrogen produced can be supplied to the hydrogen downstream industry, thus, improving the operation economy of the SCESS-DFIG-P2H generating station.

(2) Based on the primary frequency regulation mechanism of the synchronous unit, the corresponding compensation power value of the system frequency drop value is analyzed. In the SCESS-DFIG-P2H generating station, the P2H device and the energy storage device cooperate for the power compensation, so that the SCESS-DFIG-P2H generating station has the same frequency regulation effect as the synchronous unit.

(3) The selection and dynamic characteristics analysis of the P2H device and SCESS device are completed, combining the advantages of a power output of both, and a cooperative frequency regulation control strategy based on the P2H device and SCESS device is proposed. The simulation model of the SCESS-DFIG-P2H generating station is completed, and the effectiveness of the proposed control strategy in primary frequency regulation is verified through a simulation.

## 2. Research on Primary Frequency Regulation Technology Based on a SCESS-DFIG-P2H Generating Station

### 2.1. System Architecture of the SCESS-DFIG-P2H Generating Station

The system architecture of the SCESS-DFIG-P2H generating station is shown in Figure 1. The SCESS-DFIG-P2H generating station system includes a distribution network in module I, a wind turbine group in module II, the P2H system in module IV and the hydrogen consumption industry.

Among them, the wind turbine group is composed of a plurality of SCESS-DFIGs, shown in module III, as the energy source of the whole system. The mechanical power $P_{\text{wind}}$ captured by the wind turbine during the power generation is shown in Formula (1):

$$P_{\text{wind}} = \frac{1}{2}\rho\pi R^2 v^3 C_{\text{p}} \tag{1}$$

Among them:

$$\begin{cases} C_{\text{p}} = 0.02\left(\frac{116}{\lambda_{\text{i}}} - 0.4\beta - 5\right)e^{-\frac{12.5}{\lambda_{\text{i}}}} \\ \frac{1}{\lambda_{\text{i}}} = \frac{1}{\lambda + 0.08\beta} - \frac{0.035}{\beta^3 + 1} \\ \lambda = \frac{\Omega R}{v} \end{cases} \tag{2}$$

where $R$ is the radius of the wind turbine blade; $\rho$ is the air density; $v$ is the wind speed; $C_{\text{p}}$ is the wind energy utilization coefficient of the wind turbine; $\lambda$ is the tip speed ratio; $\beta$ is the blade angle; $\Omega$ is the mechanical angular velocity of the DFIG blade.

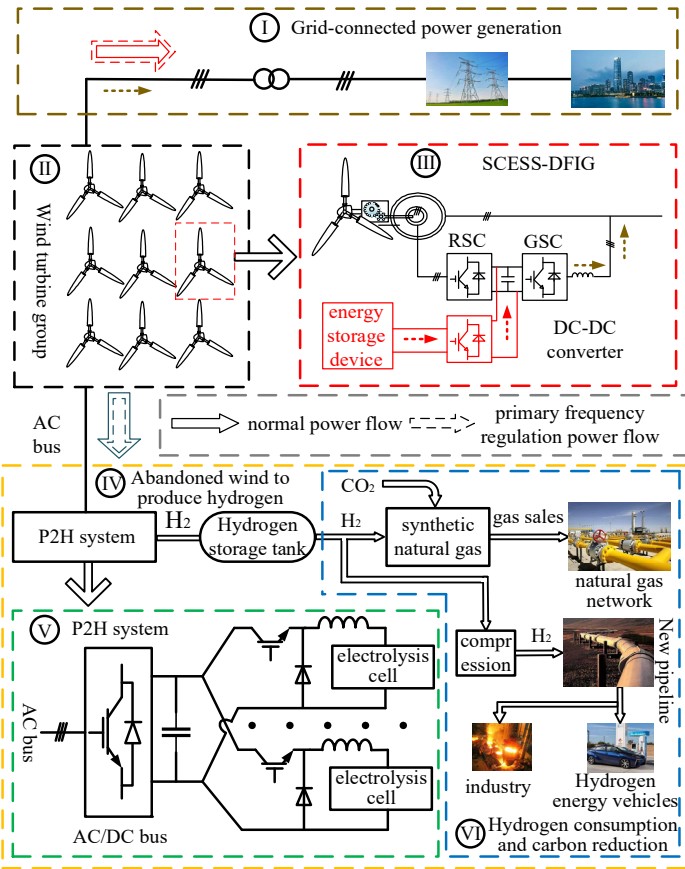

**Figure 1.** Schematic diagram of system architecture of a SCESS-DFIG-P2H generating station.

In order to make full use of wind energy, the DFIG runs normally in the MPPT state. At this time, the wind energy utilization coefficient obtains the maximum value $C_{pmax}$, so that the $P_{wind}$ obtains the maximum value $P_{MPPT}$.

The mathematical model of the DFIG in a two-phase rotating coordinate system is used for the motor control, and the voltage equation of the DFIG is obtained as follows:

$$\begin{cases} u_{sd} = -i_{sd}R_s - \omega_s\psi_{sq} + \frac{d\psi_{sd}}{dt} \\ u_{sq} = -i_{sq}R_s + \omega_s\psi_{sd} + \frac{d\psi_{sq}}{dt} \\ u_{rd} = -i_{rd}R_s - (\omega_s - \omega_m)\psi_{rq} + \frac{d\psi_{rd}}{dt} \\ u_{rq} = -i_{rq}R_s + (\omega_s - \omega_m)\psi_{rd} + \frac{d\psi_{rq}}{dt} \end{cases} \quad (3)$$

In the formula, $u_{sd}$, $u_{sq}$, $u_{rd}$, and $u_{rq}$ are the stator voltage and rotor voltage components under rotation in the d and q coordinate systems, respectively; $i_{sd}$, $i_{sq}$, $i_{rd}$, and $i_{rq}$ are the stator current and rotor current components under rotation in the d and q coordinate systems, respectively; $\psi_{sd}$, $\psi_{sq}$, $\psi_{rd}$, and $\psi_{rq}$ are the stator and rotor magnetic chain components under the rotation of the d and q coordinate systems, respectively; $\omega_s$ is the electric angular velocity of the d and q coordinate system rotation; $\omega_m$ is the electric angular velocity of the generator input.

The equation of the magnetic chain is:

$$\begin{cases} \psi_{sd} = -L_m(i_{sd} + i_{rd}) \\ \psi_{sq} = -L_m(i_{sq} + i_{rq}) \\ \psi_{rd} = -L_m i_{sd} - (L_\sigma + L_m)i_{rd} = -L_m i_{sd} - L_R i_{rd} \\ \psi_{rq} = -L_m i_{sq} - (L_\sigma + L_m)i_{rq} = -L_m i_{sq} - L_R i_{rq} \end{cases} \quad (4)$$

In the formula, $L_\sigma$ is the leakage inductance between the stator and rotor under the d and q coordinate system rotation; $L_m$ is the mutual inductance between the stator and rotor

under the d and q coordinate system rotation; $L_R$ is the sum of the leakage and mutual inductance between the stator and rotor under the d and q coordinate system rotation.

The power relationship analysis of the energy conversion system of the DFIG is shown in Figures 2 and 3.

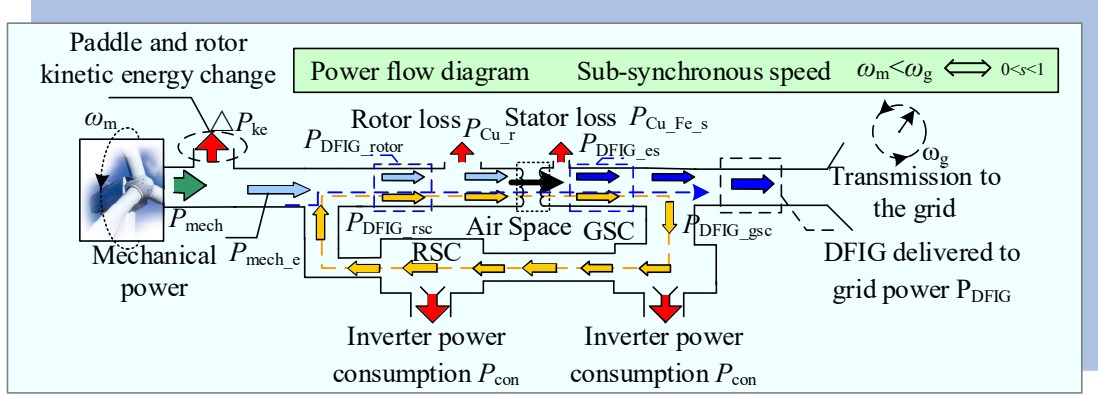

**Figure 2.** The power flow relationship of DFIG in a sub-synchronous operation.

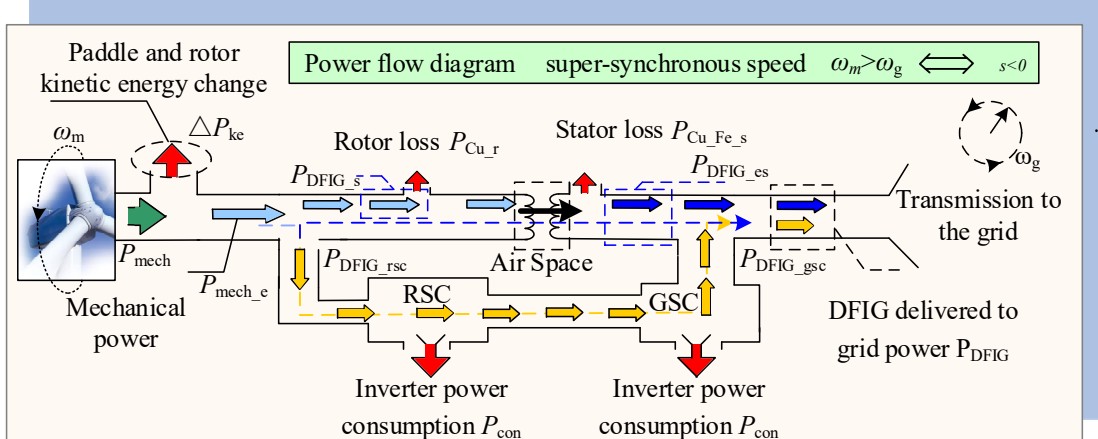

**Figure 3.** The power flow relation of the DFIG with an over-synchronous operation.

When 0 < s < 1 ($\omega_m < \omega_g$), the DFIG is in a sub-synchronous operation, and its power flow relationship is shown in Figure 2. The GSC absorbs electrical power $P_{DFIG\_gsc}$ from the system, and the power flows to the RSC to provide the AC excitation power $P_{DFIG\_rsc}$ for the DFIG excitation power $P_{DFIG\_rsc}$ and the wind turbine input generator mechanical power. The sum of $P_{mech\_e}$ flows to the stator end of the DFIG through an electromagnetic conversion, the stator output power is the $P_{DFIG\_es}$ delivered to the grid in the process, and part of the power $P_{DFIG\_gsc}$ flows to the GSC as described earlier. The actual system's instantaneous power delivered to the grid is $P_{DFIG}$. The rotor loss power is $P_{Cu\_r}$, the stator loss power is $P_{Cu\_Fe\_s}$, and the converter loss power is $P_{con}$.

When s < 0 ($\omega_m > \omega_g$), the DFIG is in over-synchronous operation and its power flow relationship is shown in Figure 3. From Figure 3, it can be seen that as in the over-synchronous operation, the mechanical power of the wind turbine input generator is $P_{mech\_e}$ which is excited by the RSC to the rotor of the DFIG, turning the mechanical energy into electrical energy. Different from the sub-synchronous operation state, in the over-synchronous operation state, the mechanical power $P_{mech\_e}$ is divided into two parts, where one part is delivered by the stator to the grid size $P_{DFIG\_es}$, the other part of the power ($P_{DFIG\_rsc}/P_{DFIG\_gsc}$) is delivered to the grid by the rotor side converter, and the GSC of the DFIG is used as a rectifier to ensure the DC bus stability; therefore, $P_{DFIG\_rsc} + P_{DFIG\_gsc} = 0$. The rotor-side converter power flow direction is the power flow from the RSC to the GSC, and the actual system instantaneous power delivered to the grid is $P_{DFIG}$.

In the system, part of the output power of the wind turbine is sent to module I for grid-connected power generation, and the other part is sent to module IV. Module IV includes module V and module VI. Module V is the P2H system, which consists of a rectifier, DC–DC converter and electrolysis cells; the downstream industry of hydrogen consumption is shown in module VI.

When the SCESS-DFIG-P2H generating station is connected to the grid for power generation, the doubly-fed wind turbine as the energy source runs in a stable state of maximum wind power tracking, and its output power is only related to the wind speed, completely decoupled from the grid frequency, and it does not have the synchronous unit response system frequency ability to change. Therefore, in order to enhance the frequency regulation capability of the SCESS-DFIG-P2H generating station system, it is necessary to analyze the SCESS-DFIG-P2H generating station according to the primary frequency regulation mechanism of a traditional, synchronous generator set, and to design a primary frequency regulation control strategy to ensure it has the same primary frequency regulation effect as the traditional synchronous unit.

### 2.2. Research on the Primary Frequency Regulation Characteristics of a SCESS-DFIG-P2H Generating Station Based on a Traditional Synchronous Generator Set

The primary frequency regulation of traditional synchronous units includes inertia response and droop characteristics. The mechanism of action of both is to increase the electromagnetic torque of the generator to provide an additional power output and to participate in the primary frequency regulation of the system. The former is the effect of the rotor speed responding to the frequency change of the system, which is a passive response, while the droop characteristic increases the active power output by the action of a speed regulating mechanism, which is an active adjustment. The schematic diagram of the shaft system of the synchronous unit is shown in Figure 4. The prime mover produces the mechanical torque $T_M$, and the generator changes the output power of the synchronous unit through the electromagnetic torque $T_E$. The inertia response process suppresses the frequency change rate of the system by generating the electromagnetic torque increment $\Delta T_{E_1}$. The droop characteristic is acted upon by the governor to generate the electromagnetic torque increment $\Delta T_{E_2}$ to reduce a frequency change in the system. The mechanism analysis process of the inertia response and droop characteristics is detailed in Appendix A.

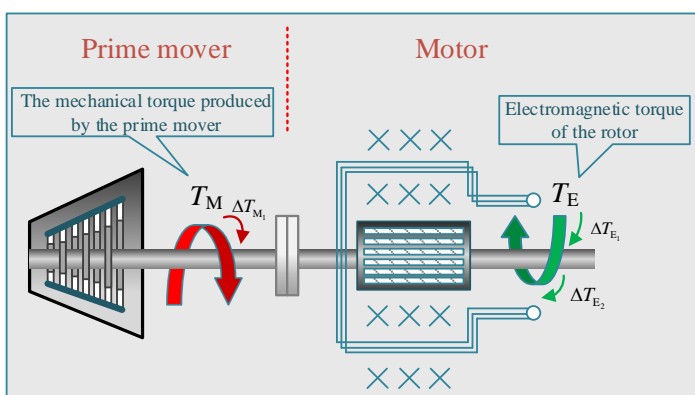

**Figure 4.** Shafting diagram of a synchronous set.

Synchronous generator sets generally use an inertia time constant $T_J$ to characterize the influence of the generator set inertia on its dynamic behavior. The expression is shown in Formula (5):

$$T_J = \frac{2E_{ks}}{S_N} = \frac{J\omega_S^2}{S_N} \tag{5}$$

In the formula, $E_{ks}$ is the rotor kinetic energy of the generator at the rated speed; $J$ is the moment of inertia of the generator; $\omega_S$ is the rated speed of the generator; $S_N$ is the rated capacity of the generator.

In the SCESS-DFIG-P2H generating station system, the wind turbine, the energy storage device and the P2H device, which can be regarded as a controllable load, are used to change the power flow of the system and complete the primary frequency regulation service similar to the synchronous unit. Analogous to Formula (5), the simulated inertia of the SCESS-DFIG-P2H generating station system can be obtained:

$$H_{\mathrm{WSP}} = \frac{\left[ \sum\limits_{i=1}^{n} E_{\mathrm{w.i}} + E_{\mathrm{S}} + E_{\mathrm{P}} \right]}{S_{\mathrm{NW}}} \tag{6}$$

In the formula, $\sum\limits_{i=1}^{n} E_{\mathrm{w.i}} = 0$ is the sum of the blade kinetic energy of n wind turbines in the SCESS-DFIG-P2H generating station system; $E_S$ is the energy that can be handled by the energy storage device; $E_P$ is the energy provided by the electric hydrogen production device as a controllable load; $S_{NW}$ is the wind farm rated capacity.

Formula (6) only shows that the SCESS-DFIG-P2H generating station system has the ability to generate an inertia constant of $H_{WSP}$, but when the system frequency changes, it does not necessarily show the inertia characteristics; therefore, it is necessary to carry out research from the dynamic characteristics of the system.

Assuming that the mechanical power of the synchronous unit is constant, the per-unit value expression of the inertia constant can be obtained from the equation of motion of the rotor:

$$T_{\mathrm{J}} = \frac{\Delta P_{\mathrm{e}}^*}{\omega^* \frac{d\omega^*}{dt}} = \frac{\Delta T_{\mathrm{e}}^*}{\frac{d\omega^*}{dt}} = 2H_{\mathrm{WSP}} \tag{7}$$

In the formula, $\Delta P_{\mathrm{e}}^*$ is the per-unit value of the generator electromagnetic change power; $\omega^*$ is the per-unit value of the rotor speed; $\Delta T_{\mathrm{e}}^*$ is the per-unit value of the generator electromagnetic torque change. Referring to the definition of the inertia constant of the synchronous unit in Equation (7), the generalized inertia constant of the SCESS-DFIG-P2H generating station system can be defined: when the system frequency changes, the electromagnetic torque increment caused by the energy storage element and the P2H device is equal to the 1/2 ratio of the rotor speed change rate of the effective generator that is used as the inertia constant $H_{WSP}$, which is used to characterize the ability of the SCESS-DFIG-P2H generating station system to respond to frequency changes.

Since the per-unit values of the system frequency and the rotor speed of the equivalent synchronous generator are the same, Equation (7) can be rewritten as:

$$f^* \frac{df^*}{dt} H_{\mathrm{WSP}} = \frac{\Delta P_{\mathrm{e}}^*}{2} \tag{8}$$

Among them, $f^*$ is the system frequency per-unit value. Both sides of Equation (8) are integrated with time to obtain the average generalized inertia constant of the SCESS-DFIG-P2H generating station system in a short period of time:

$$H_{\mathrm{WSP}} = \frac{\Delta E^*}{f^{*2}(t + \Delta t) - f^{*2}(t)} \tag{9}$$

Among them, $f^*(t)$ is the per-unit value of the frequency of the system at time $t$; $f^*(t + \Delta t)$ is the per-unit value of the frequency at time $t + \Delta t$; $\Delta E^*$ is the per-unit value of the energy that can be released by the energy storage device and P2H system in time. From Equation (9), it can be seen that the SCESS-DFIG-P2H generating station provides energy support by controlling the energy storage device and the P2H system, and it responds to the frequency change of the system, showing inertia characteristics.

### 3. Analysis of the Operating Mechanism of the SCESS-DFIG-P2H Generating Station System Based on Primary Frequency Regulation

In the above, through the analysis of the traditional synchronous unit, the average generalized inertia constant of the SCESS-DFIG-P2H generating station system was defined, and the feasibility of the system with a frequency regulation capability by changing the power output was analyzed. This chapter will conduct research from the perspective of the power required to respond to frequency regulation, and analyze the operating characteristics of a P2H device and energy storage device involved in frequency regulation.

*3.1. The Capacity Matching Design of the Wind Storage Hydrogen Production Power Station Participating in the Primary Frequency Regulation of the System*

When responding to the system frequency change, the rotor speed of a synchronous unit has a coupling relationship with the grid frequency, limiting the frequency change to a small range (i.e., the domestic grid guidelines [7] require that the grid frequency fluctuates within 47 Hz to 52 Hz).

Since the rise and fall of the system frequency are similar, and the rising degree of the frequency is smaller than the falling degree of the frequency, only the downward fluctuation of the frequency will be analyzed below. The energy source of the inertia response energy of a synchronous unit is the rotor kinetic energy. Assuming that the grid frequency drops from 50 Hz to $f_1$, the rotational speed of the synchronous unit during the frequency modulation process changes as $f_1/50$~1 pu, and the rotor kinetic energy released by the synchronous unit is:

$$\Delta E_{\text{k\_max}} = \frac{1}{2} J \left( 1^2 - (f_1/50)^2 \right) \omega_\text{S}^2 \tag{10}$$

According to the law of energy conservation, if the SCESS-DFIG-P2H generating station has the same inertia response characteristics as a traditional power station, the energy released during the inertia response process should be equivalent to that of the synchronous unit:

$$\Delta E_\text{W} = \Delta E_{\text{k\_max}} = P_\text{W} \cdot \Delta t = \frac{2500 - f_1^2}{5000} P_\text{N} T_\text{J} \tag{11}$$

In the formula, $P_\text{N}$ is the rated power of the generator. Assuming that the time when the SCESS-DFIG-P2H generating station participates in the inertia response of the system is consistent with the inertia time constant of the synchronous generator set, that is $T_\text{J} = \Delta t$, the power required by the SCESS-DFIG-P2H generating station to complete the inertia response is:

$$P_\text{W} = \frac{2500 - f_1^2}{5000} P_\text{N} \tag{12}$$

From the power frequency characteristic curve of the synchronous unit, when the system frequency drops to $f_1$, the power support provided by the droop characteristic is:

$$P_\text{V} = K \cdot (50 - f_1) \tag{13}$$

In the formula, $K$ is the droop coefficient of the generator. To sum up, if the SCESS-DFIG-P2H generating station has the same primary frequency regulation effect as the traditional power station, the total amount of the frequency regulation power that needs to be provided is:

$$P_\text{WSP} = \frac{2500 - f_1^2}{5000} P_\text{N} + K \cdot (50 - f_1) \tag{14}$$

As shown in Formula (14), the total power required by the SCESS-DFIG-P2H generating station to complete one frequency regulation is $P_\text{WSP}$, which is shared by the energy storage device and the P2H device. In order to allocate the frequency regulation power reasonably, the mathematical model and dynamic characteristics of the P2H device are firstly analyzed.

### 3.2. Mathematical Model and Dynamic Characteristics of a P2H Device

At present, there are three main types of electrolysis cells, including an alkaline electrolysis cell, a proton exchange membrane electrolysis cell, and a solid oxide electrolysis cell. An alkaline electrolysis cell has the advantages of having the earliest research and development, the most mature technology, and the lowest equipment cost. In this paper, the alkaline electrolytic cell was selected, and the electrolytic cell model was established as follows.

The voltage and current equations of the single cell of the electrolytic cell are shown in Formula (15):

$$
\begin{aligned}
U_{\text{cell}} \quad &= U_{\text{rev}} + \frac{r_1 + r_2 T_{\text{el}}}{A_{\text{el}}} I_{\text{el}} + \left(s_1 + s_2 T_{\text{el}} + s_3 T_{\text{el}}^2\right) \cdot \\
&\quad \lg\left(\frac{t_1 + \frac{t_2}{T_{\text{el}}} + \frac{t_3}{T_{\text{el}}^2}}{A_{\text{el}}} I_{\text{el}} + 1\right)
\end{aligned}
\tag{15}
$$

In the formula, $U_{\text{cell}}$ is the single cell voltage of the electrolytic cell; $I_{\text{el}}$ is the current flowing through the electrolytic cell; $U_{\text{rev}}$ is the reversible cell voltage; $r_1$, and $r_2$ are the ohmic parameters of the electrolyte; $s_1$, $s_2$, $s_3$, $t_1$, $t_2$, and $t_3$ are the electrodes overvoltage parameters; $A_{\text{el}}$ is the electrode area; $T_{\text{el}}$ is the electrolyte temperature.

The series voltage equation of the electrolysis cells is shown in Formula (16):

$$
U_{\text{el}} = N_{\text{el}} U_{\text{cell}}
\tag{16}
$$

In the formula, $N_{\text{el}}$ is the number of batteries connected in series in the electrolytic cell; $U_{\text{el}}$ is the voltage of the electrolytic cell.

According to Faraday's law, the hydrogen production rate of an electrolytic cell is proportional to the equivalent current $I_{\text{el}}$ of the electrolytic cell, and the expression is shown in Equation (17):

$$
\begin{cases}
V_{\text{H}_2} = \frac{\eta_F N_{\text{el}} I_{\text{el}}}{2F} \\
\eta_F = 96.5 \mathrm{e}^{(0.09/I_{\text{el}} - 75.5/I_{\text{el}}^2)}
\end{cases}
\tag{17}
$$

In the formula, $\eta_F$ is the faradaic efficiency; $V_{\text{H}_2}$ is the hydrogen production rate of the electrolysis cells.

Considering the influence of the working characteristics of the alkaline electrolytic cell on the response to the frequency modulation power command, the main technical parameters of the alkaline electrolytic cell were selected for analysis [34], as shown in Table 1:

**Table 1.** Main technical indexes of an alkaline electrolytic cell.

| | Main Technical Indicators | Indicators Value |
|---|---|---|
| Start-stop | Start-stop delay $\alpha$/h | 0 |
| | Start-stop capability $Y^{\text{max}}/Z^{\text{max}}$ | 1/1 |
| Power | Work scope $[P^{\text{min}}, P^{\text{max}}]$ | $[25\% P_{\text{el}}^{\text{max}}, 100\% P_{\text{el}}^{\text{max}}]$ |
| | Gradeability $\Delta P^{\text{max}}$ | $<(1/7200)P_{\text{el}}^{\text{max}}/\text{s}$ |

From Table 1, it can be concluded that an alkaline electrolytic cell has no startup delay, and can start up quickly to consume electricity and produce hydrogen. Considering that the working range of an alkaline electrolytic cell is 25% to 100% of the rated power $P_{\text{el}}^{\text{max}}$, and only a single start and stop can be performed in a single day, the power command constraint for the electrolysis cells participating in the primary frequency modulation of the system is shown in Formula (18):

$$
\frac{1}{4} N_{\text{el}} P_{\text{el}}^{\text{max}} \leq P_{\text{P2H}} \leq N_{\text{el}} P_{\text{el}}^{\text{max}}
\tag{18}
$$

In addition, limited by the inertia of temperature and material changes, an alkaline electrolysis cell needs to meet the limit of the power ramping command during operation:

$$|P_{\mathrm{el,t}} - P_{\mathrm{el,t-1}}| \le \Delta P^{\max} \tag{19}$$

In the formula, $P_{\mathrm{el,t}}$ is the power command that the electrolytic cell needs to undertake at time $t$; $\Delta P^{\max}$ is the power change threshold per unit time.

To sum up, an alkaline electrolyzer is limited by its own working characteristics, and there is a difference between the response frequency modulation power and the total frequency modulation power. An energy storage device can undertake the power command beyond the working range of a P2H device through a fast energy throughput, to ensure that the system responds to frequency regulation power in real time, to enhance the frequency regulation capability of the system, and to improve the stability of the system operation.

### 3.3. Comparative Analysis of the Electrical Characteristics of Different Energy Storage Devices Combined with a P2H Device

The main technical parameters of each energy storage device were selected [35], and both the electrical as well as the economic characteristics of each energy storage device and the combination of P2H were contrastively analyzed, as shown in Figure 5.

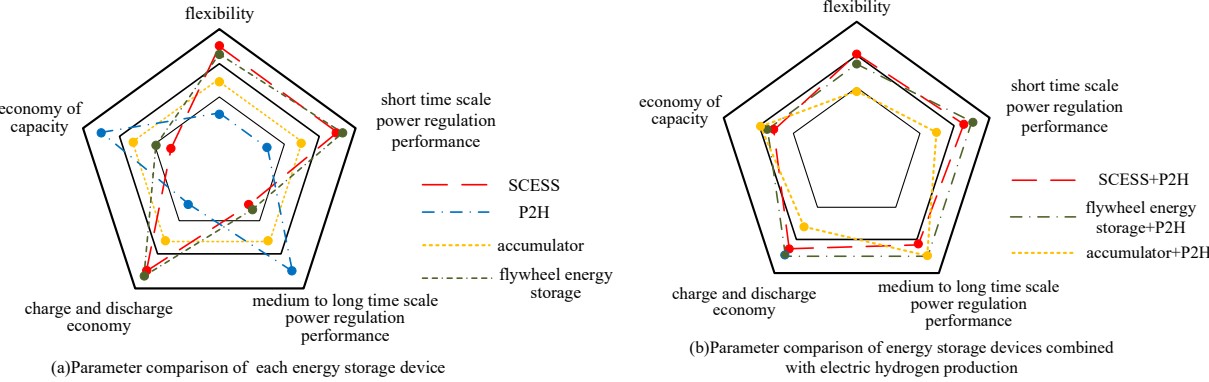

**Figure 5.** Comparison of parameters for various energy storage devices and combined with P2H.

As shown in Figure 5a, the SCESS and flywheel energy storage can make up for the shortcomings of a P2H device in terms of flexibility and a high power output, and the lack of a medium- and long-term power regulation performance can be compensated for by the P2H device. As shown in Figure 5b, the characteristic curves of the two combined with a P2H device are closer to the outer edge of the tortoise-back diagram, and have better electrical and economic characteristics. Compared with the SCESS, the self-discharge rate of the flywheel energy storage is high, but it has not yet been developed to the stage of engineering applications; therefore, it is difficult to purchase.

In summary, the selection of a SCESS for power compensation solves the problem of limited power adjustment capabilities of a P2H device participating in a system frequency regulation service. At the same time, since the SCESS would only be used as an auxiliary device to solve the problems caused by the characteristics of a P2H device itself, the frequency regulation service would still mainly be based on the P2H; therefore, it would not require an excessive capacity configuration, and the one-time investment cost would be low.

## 4. Primary Frequency Regulation Control Based on SCESS-DFIG-P2H Generating Station

On the basis of the previous chapter, this chapter establishes the control equations of the P2H device and the SCESS, it researches the control strategies of both, and proposes a multi-energy coordinated frequency regulation control strategy for the SCESS-DFIG-P2H generating station.

### 4.1. Research on the Control Strategy of the P2H Device

The equation of the control unit of the P2H device is:

$$D_{\text{el}} = \frac{\left(K_{\text{p1}} + \frac{K_{\text{p2}}}{s}\right)(I_{\text{elref}} - I_{\text{el}}) + U_{\text{el}}}{U_{\text{eldc}}} \tag{20}$$

In the formula, $D_{\text{el}}$ is the control signal of the P2H device; $K_{\text{p1}}$ and $K_{\text{p2}}$ are the proportional adjustment gain and integral adjustment gain of the current loop, respectively; $I_{\text{elref}}$ and $I_{\text{el}}$ are the reference value and feedback value of the electrolysis cell current, respectively; $U_{\text{elc}}$ is the voltage value of the DC bus of the P2H system.

In summary, considering the working characteristics of the P2H device itself, combined with the control unit equation of the P2H device, the control strategy for establishing the P2H device is: the total power required for the primary frequency regulation of the SCESS-DFIG-P2H generating station is $P_{\text{WSP}}$, and when the load shedding responds to the power command, it is limited by the change in temperature and the material inertia and it has a response delay characteristic. The reference power value of the P2H device, therefore, is obtained through the delay link, and the limiter enables the P2H device to participate in the system frequency regulation within the working range. The power command that the electric hydrogen production device needs to undertake is $P_{\text{P2H}}$, and the reference current command $I_{\text{elref}}$ is generated by the actual voltage $U_{\text{el}}$ at both ends of the electrolytic cell. The difference between $I_{\text{elref}}$ and the actual current $I_{\text{el}}$ is controlled by PI to obtain the control signal $D_{\text{el}}$ of the P2H device. The PWM control signal is generated by the comparator, and the control block diagram of the P2H device is shown in Figure 6:

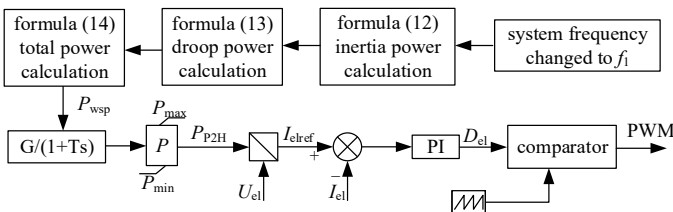

**Figure 6.** The control diagram of P2H.

The delay link transfer function mentioned above is:

$$G(s) = \frac{1}{\tau s + 1} \tag{21}$$

In the formula, $\tau$ is the time constant.

When the input is a step load, the input signal $R(s)$ satisfies:

$$R(s) = \frac{D}{s} \tag{22}$$

In the formula, $D$ is the step amplitude.

Then, its step response $C(s)$ is:

$$C(s) = \frac{D}{s} \cdot \frac{1}{\tau s + 1} + \frac{1}{\tau s + 1} \cdot \frac{[c(0) - r(0)]}{s} \tag{23}$$

In the formula, $c(0)$ and $r(0)$ represent the initial value of the state in the system and the initial value of the input, respectively.

An inverse Laplace transform is performed on Equation (23) to obtain the time domain expression:

$$C(t) = \left(1 - e^{-\frac{t}{\tau}}\right)[D + C(0) - r(0)] \tag{24}$$

According to Formula (24), the step response of the delay element is the inverse form of the negative exponent of $e$. When the initial value is 0, if a step change of the power command occurs after the calculation of the delay link, the current reference value change curve, with an initial amount equal to the change amount of the power command and attenuated according to the inverse number curve of the negative exponential function of $e$, can be obtained.

Figure 7 depicts the comparison of the frequency regulation power command with its output signal through the delay link for different time constants (1, 5, and 10).

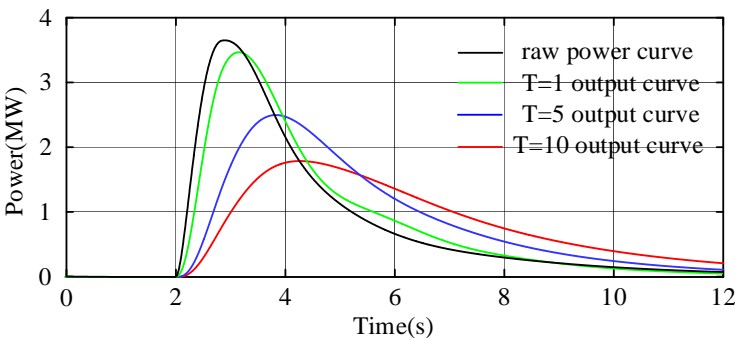

**Figure 7.** Comparison of power command filtering effect under different time constants.

As shown in Figure 7, the signal passing through the delay link becomes smoother with the increase of the time constant, but the curve delay also increases; therefore, under different working conditions, the P2H device should select a time constant that meets the corresponding requirements. The expression of the time constant $T$ is established as follows:

$$T = k \cdot \frac{1}{N} + \sum_{i=1}^{N} k_i \frac{1}{P_i} \tag{25}$$

In the formula, $N$ is the number of electrolytic cells opened; $P_i$ is the operation power of the electrolytic cells; $k$ is the correlation coefficient between $T$ and $1/N$; $k_i$ is the correlation coefficient between $T$ and $1/P_i$.

### 4.2. Research on the Control Strategy of a Supercapacitor Energy Storage Device

The SCESS is coupled to the DC bus of the energy storage DFIG through a bidirectional DC–DC converter. The DC–DC converter operating modes are divided into a Buck mode and Boost mode.

When the bidirectional DC–DC converter is in the Buck mode, the control equation is:

$$D_{\text{scbuck}} = \frac{\left(K_{s1} + \frac{K_{s2}}{s}\right)(I_{\text{scref}} - I_{\text{sc}}) + U_{\text{sc}}}{U_{\text{dc}}} \tag{26}$$

In the formula, $D_{\text{scbuck}}$ is the control signal of the bidirectional DC–DC converter of the SCESS in Buck mode; $K_{s1}$ and $K_{s2}$ are the proportional adjustment gain and integral adjustment gain of the current loop, respectively; $I_{\text{scref}}$ and $I_{\text{sc}}$ are the reference and feedback values of the current flowing through the SCESS, respectively; $U_{\text{sc}}$ is the voltage value across the SCESS; $U_{\text{dc}}$ is the DC bus voltage value of the DFIG.

When the bidirectional DC–DC converter is in the Boost mode, the control equation is:

$$D_{\text{scboost}} = \frac{\left(K_{s1} + \frac{K_{s2}}{s}\right)(I_{\text{scref}} - I_{\text{sc}}) + U_{\text{sc}} - U_{\text{dc}}}{\left(K_{s1} + \frac{K_{s2}}{s}\right)(I_{\text{scref}} - I_{\text{sc}}) + U_{\text{sc}}} \tag{27}$$

In the formula, $D_{\text{scboost}}$ is the control signal when the bidirectional DC–DC converter of the SCESS is in Boost mode.

The SCESS adopts a constant power control strategy as shown in Figure 8. The SCESS is rapidly charged and discharged to absorb/compensate for the response delay of the P2H device and the difference in the frequency regulation power caused by a power limitation (the difference between $P_{\text{WSP}}$ and $P_{\text{P2H}}$). The SCESS needs to undertake the quotient between the power command $P_{\text{sc}}$ and the actual voltage $U_{\text{sc}}$ at both ends of the SCESS to generate the reference current command $I_{\text{scref}}$. The difference between $I_{\text{scref}}$ and the actual current $I_{\text{sc}}$ is controlled by the PI to generate the control signal $D_{\text{sc}}$, and the triangular

wave is passed through the comparator to generate the PWM control signal, which acts on switching devices for bidirectional DC–DC converters.

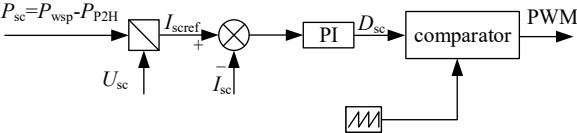

**Figure 8.** Control diagram of SCESS.

### 4.3. Primary Frequency Regulation Control Strategy of a SCESS-DFIG-P2H Generating Station

Combined with the above-mentioned control strategies for a P2H device and a SCESS, a multi-energy collaborative control strategy for wind-storage-hydrogen power stations is established. The overall control block diagram is shown in Figure 9.

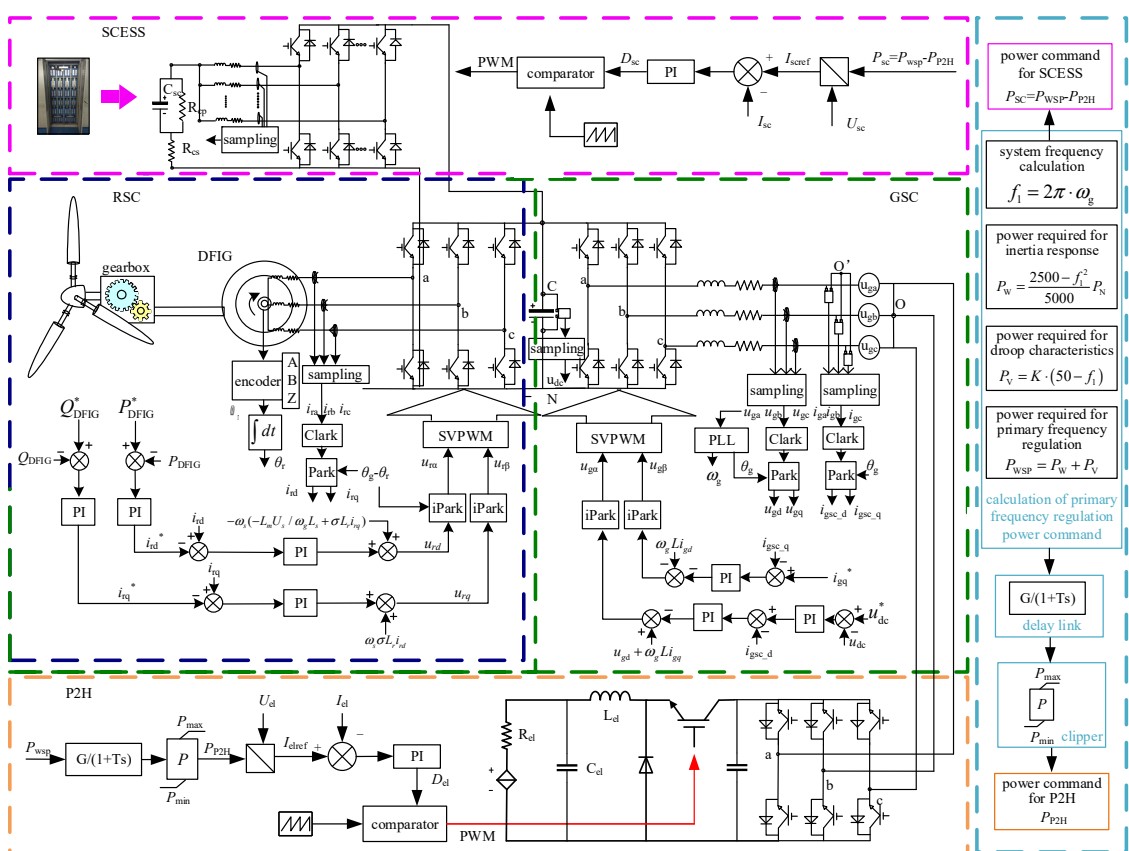

**Figure 9.** General control block diagram of wind storage hydrogen power station.

As shown in Figure 9, the wind energy captured by a wind turbine blade is converted into the kinetic energy of the blade, and is connected to the shafting of a doubly-fed induction wind turbine through a gearbox. The dual PWM converter is composed of GSC and RSC. The GSC adopts the current inner loop and the voltage outer loop control mode to keep the DC bus voltage constant. The RSC adopts the control mode of the current inner loop and power outer loop to adjust the active output of the DFIG. The grid side has three-phase voltage and current Hall sensors for real-time sampling. The collected three-phase grid voltage is locked to the angular frequency of the grid through the phase-locked loop, and the inertia response and the power command required for the droop characteristic are obtained by the calculation link. The power command obtains the power command that a P2H device needs to undertake through the delay link and the power limiter. The SCESS compensates for the difference of the frequency regulation power caused by the response delay of the P2H device, and at the same time undertakes the frequency

regulation power beyond the working range of the P2H device instruction. For the control link of the P2H device and the SCESS, the power command is divided by the real-time feedback voltage value to obtain the current feedback value. The difference between the feedback value and the actual current value is obtained through the PI link to obtain the switching signal, which is compared with the triangular wave to generate a PWM wave, which acts on the respective power switching devices to realize the power control of both, and it realizes the power response of the primary frequency regulation of the participating system; therefore, the SCESS-DFIG-P2H generating station has the same primary frequency regulation capability as a synchronous unit.

The overall control flow chart of the SCESS-DFIG-P2H generating station system is shown in Figure 10. When the system frequency changes, the phase-locked loop first feeds back the system frequency change value. The dead zone limit setting for the system frequency adjustment is 0.03 Hz. When the frequency change exceeds the set value, the control system will act. The analysis of the system frequency drop is as follows: when the system frequency drops, the original power balance is broken, and the SCESS-DFIG-P2H generating station needs to provide an additional active power output; therefore, the P2H device is used as a controllable load for load shedding, and the SCESS releases energy to complete the primary frequency regulation of the system. The power required by the SCESS-DFIG-P2H generating station simulating the inertia response of a traditional synchronous unit can be calculated by Formula (12), and the power required by the droop characteristic is calculated by Formula (13). The sum of the two powers is the total frequency regulation power. The power command passes through the delay link and the limiter to obtain the power command $P_{P2H}$ for load shedding of the P2H device, while the SCESS compensates the power difference ($P_{SC} = P_{WSP} - P_{P2H}$). The SCESS and the P2H device undertake the corresponding power command to provide an additional power output through their own control links, to assist the system to rebuild the power balance, to restore the frequency and to complete the primary frequency adjustment of the SCESS-DFIG-P2H generating station. The control mechanism when the frequency rises is shown in Figure 10.

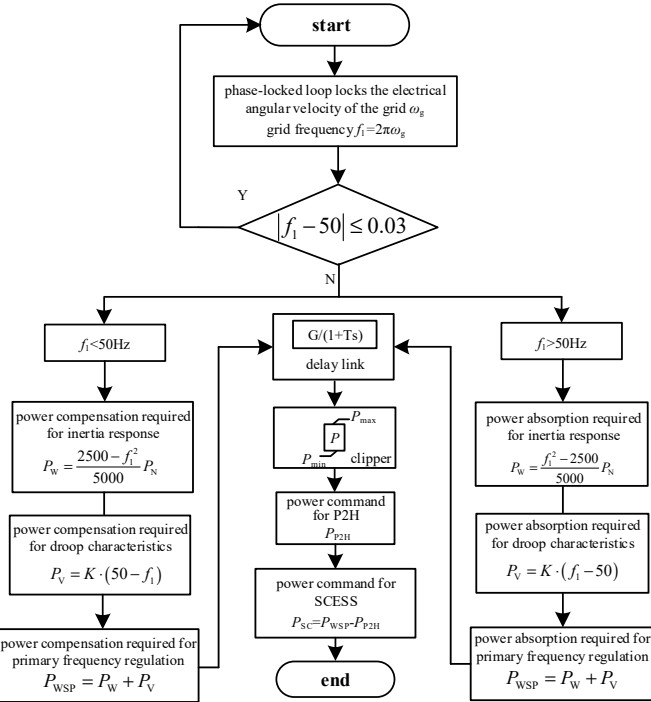

**Figure 10.** Control flow chart of wind storage hydrogen power station.

## 5. Verification of the Simulation

The simulation model of the SCESS-DFIG-P2H generating station system was built in the simulink simulation software, and its structure is shown in Figure 11. The simulation

model included a wind farm composed of three groups of energy storage types, a DFIG with 7.5 MW of rated power, a synchronous generator SG, load $L_1$, and load $L_2$.

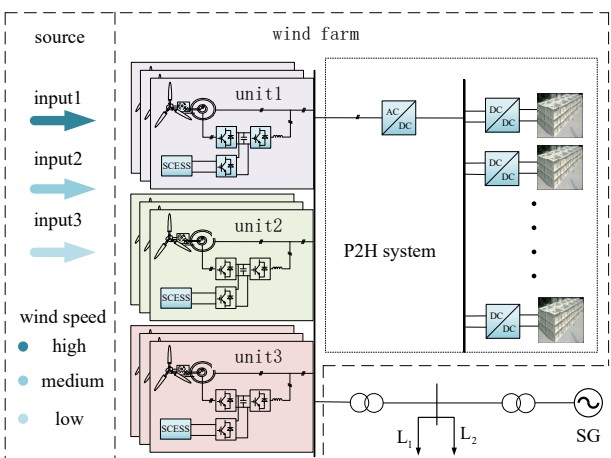

**Figure 11.** Simulation model of a SCESS-DFIG-P2H generating station.

As shown in Figure 11, the three units in the wind farm were connected to the 17 MW P2H system consisting of 10 groups of 1.7 MW P2H modules through the AC bus at different wind speeds. The wind farm was connected to the loads $L_1$, $L_2$ and the synchronous generator SG through a step-up transformer. Among them, the rated power of the synchronous generator was 100 MW, equipped with an excitation regulator and governor, the $L_1$ was a fixed load of 30 MW, and the $L_2$ was a sudden load that caused the system frequency to drop. The simulation parameters of the wind turbine and synchronous generator are shown in Appendix B (Table A1).

When the wind speed was 7 m/s, the output power of the DFIG was 0.53 MW, and the wind turbine did not meet the grid-connected power generation conditions. When the wind speed was 12 m/s, the output power of the DFIG was rated 2.5 MW, and the output power of the fan did not change with the increase in the wind speed. As a result, by comparing the output power of the DFIG, this paper defines the low wind speed as 8 m/s, the medium wind speed as 10 m/s, and the high wind speed as 12 m/s.

The three groups of SCESS-DFIGs were all set to work at a medium wind speed with a wind speed of 10 m/s. The proposed control strategy was adopted, and the system frequency regulation curve, P2H and SCESS operation state are shown in Figures 12–14 with different time constants of the delay link.

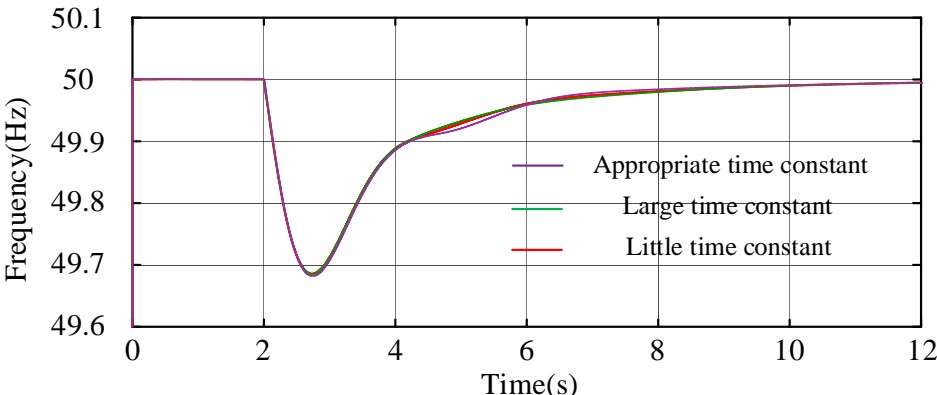

**Figure 12.** System frequency regulation result with different time constants of the delay link.

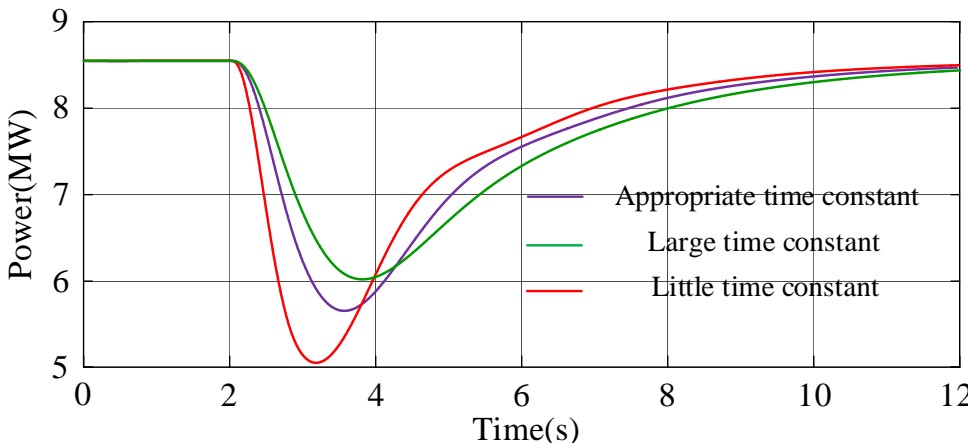

**Figure 13.** Operation curve of P2H with different time constants of the delay link.

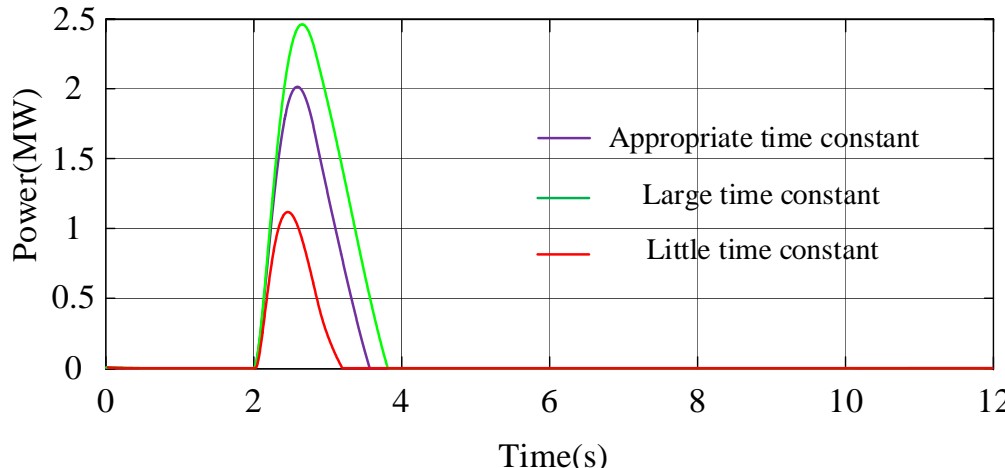

**Figure 14.** Operation curve of SCESS with different time constants of the delay link.

As shown in Figure 12, when the proposed control strategy was adopted and different time constants were used in the delay link, the system frequency deviation was 0.32 Hz; however, when the time constant was suitable, too small and too large, the times for the system frequency to recover to reasonable intervals was different, which were 6.39 s, 6.575 s and 6.68 s respectively. As shown in Figures 13 and 14, with the different time constants that led the P2H device and the SCESS device to their power allocation, the appropriate time constant could cause the P2H load shedding response to have a long time-scale frequency regulation and the SCESS to rapidly release energy in response to a short time-scale of frequency regulation, while the system could achieve a better frequency regulation combined with the power output characteristics of both.

In the following, the proposed control strategy is adopted to select the optimal time constant to simulate and analyze the system frequency regulation results under different wind speed states, while also considering the aggregation effect of wind power plants.

### 5.1. Simulation Analysis at a Low Wind Speed

To verify the effectiveness of the proposed strategy, three groups of SCESS-DFIGs were set to work in a low wind speed state, and the wind speed was 8 m/s. The surge load was added at 2 s, and its values were 10 MW, 15 MW and 20 MW, respectively. Figures 15–18 show the system frequency regulation curve, the P2H and SCESS operating curves, and the GSC and RSC running status curves of the DFIG.

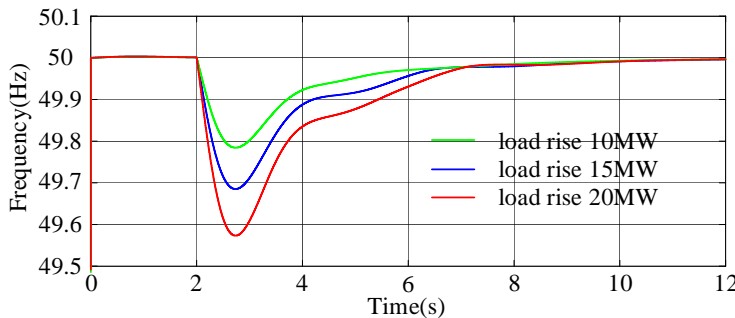

**Figure 15.** System frequency regulation result of low wind speed.

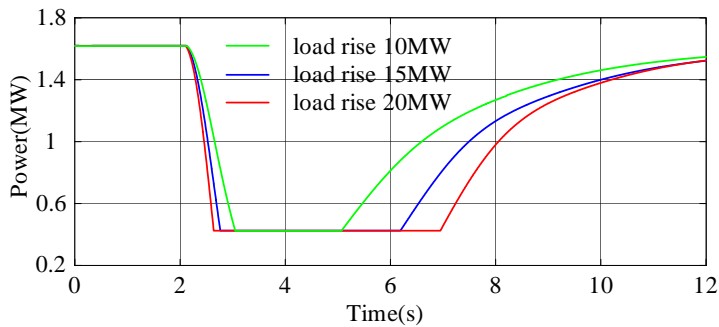

**Figure 16.** Operation curve of P2H at low wind speed.

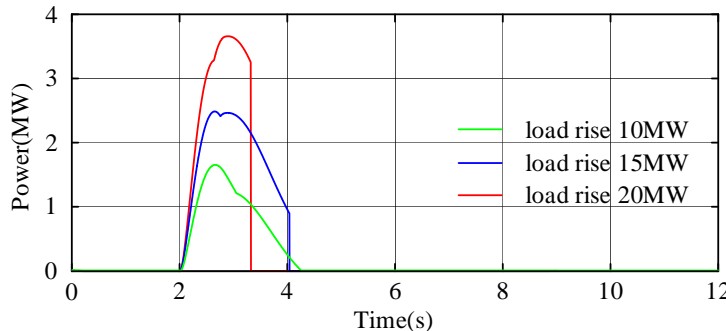

**Figure 17.** Operation curve of SCESS at low wind speed.

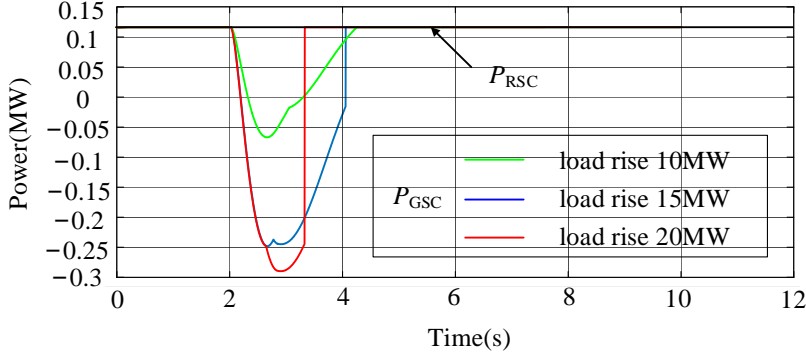

**Figure 18.** GSC and RSC running status curves of DFIG at low wind speed.

As shown in Figures 15–17, when the wind speed was low, most of the output power of the wind turbine was used for grid-connected power generation, and less power flowed to the P2H system. At this time, only one group of P2H devices was turned on, the load shedding of the P2H device was ensured within the working range (0.475 MW~1.7 MW) at the same time, and the load shedding power provided by the P2H device was limited. In order to ensure a favorable frequency regulation effect, the SCESS needed to provide

more frequency regulation power. When the surge load was 10 MW, the system frequency deviation was 0.22 Hz, and the SCESS could track the compensation power command; when the surge load was 15 MW, the system frequency deviation was 0.32 Hz, and the SCESS was discharged from the frequency regulation due to the capacity limitation in 2.77 s; when the surge load was 20 MW, the system frequency deviation was 0.43 Hz, and the SCESS quit the frequency regulation in 2.64 s. When the wind speed was low, the system had a favorable frequency regulation effect, but the energy release of the SCESS reached the upper limit and could not participate in the next frequency regulation work. As shown in Figure 18, the wind turbine ran in a sub-synchronous operation state at this time, the slip rate *s* was greater than 0, the power flowing through the RSC was 0.12 MW, and the power flowing through the GSC was the difference between the SCESS power released and itself. Since the SCESS was connected to the DC bus in parallel and exchanged energy with the power grid via the GSC, the output of the SCESS varied with different surges of load, resulting in different power flows through the GSC.

### 5.2. Simulation Analysis at a Medium Wind Speed

All three groups of the SCESS-DFIGs were set to work in the medium wind speed state, and the wind speed was 10 m/s. The surge load was added at 2 s, and its values were 10 MW, 15 MW and 20 MW, respectively. The system frequency regulation curve, the P2H device and SCESS operating curves, and the GSC and RSC running status curves of the DFIG are shown in the Figures 19–22.

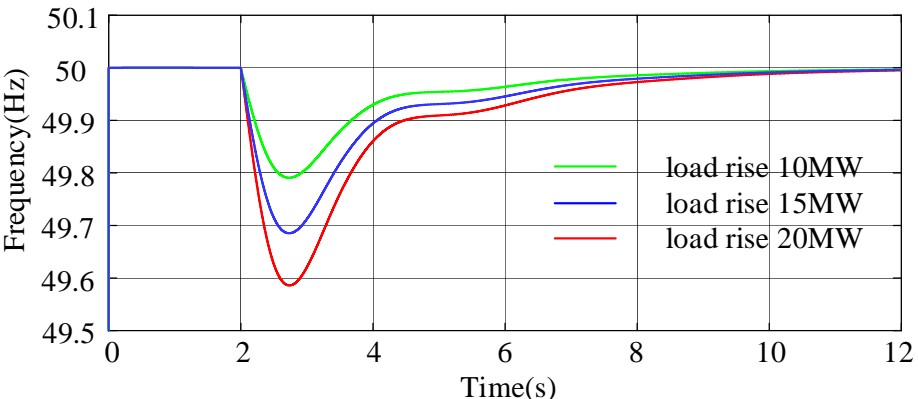

**Figure 19.** System frequency regulation result of medium wind speed.

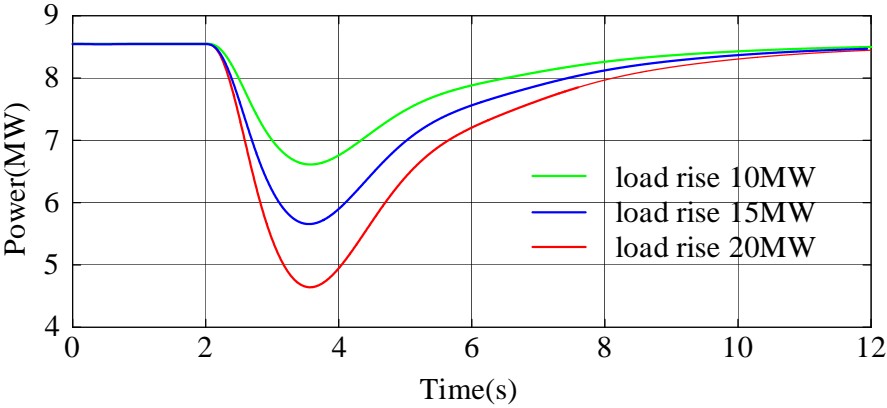

**Figure 20.** Operation curve of P2H at medium wind speed.

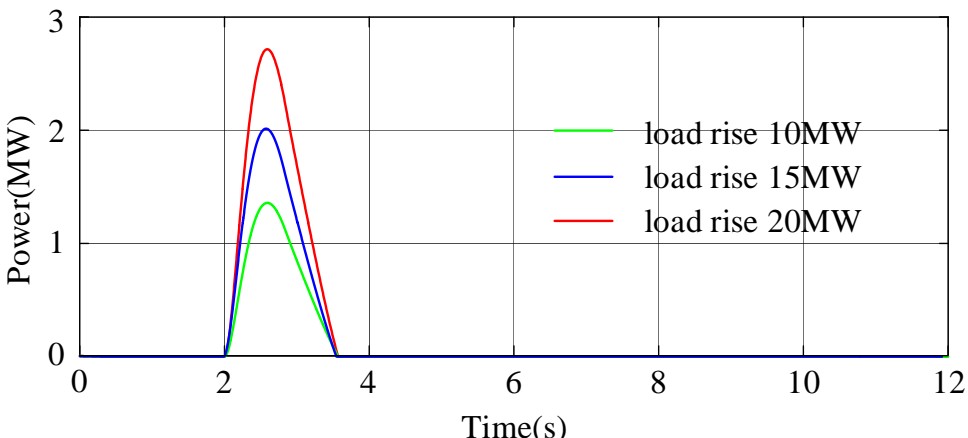

**Figure 21.** Operation curve of SCESS at medium wind speed.

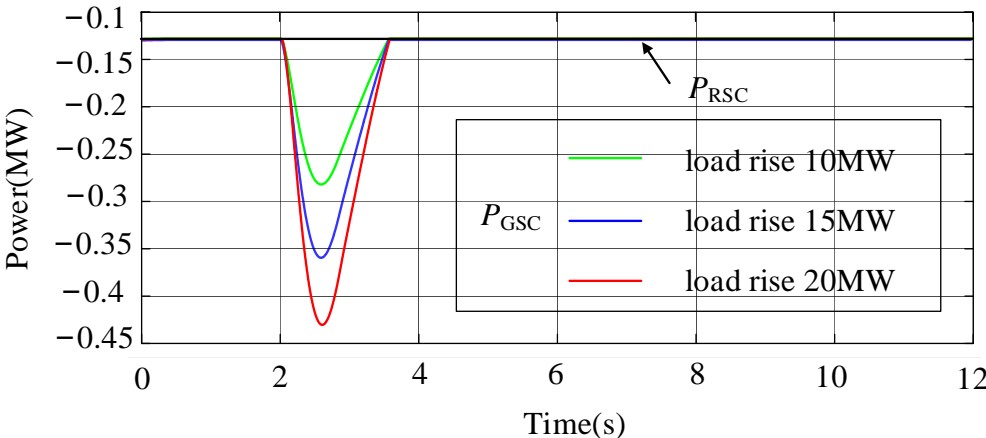

**Figure 22.** GSC and RSC running status curves of DFIG at medium wind speed.

As shown in Figures 19–21, compared with the low wind speed, the wind turbine output power increased at a medium wind speed, the initial operating power of the P2H increased, and the load shedding capacity became stronger. In the process of frequency regulation, a P2H device is the main body of the power response, and a SCESS only needs to compensate for the difference that the P2H device cannot track the frequency regulation command. Here, in the medium wind speed state, the system frequency deviations under the different surge loads were 0.21 Hz, 0.32 Hz and 0.42 Hz, respectively, which had a favorable frequency regulation effect. As shown in Figure 22, the wind turbine ran in an over-synchronous operation state at this time, the slip rate s was less than 0, the power flowing through the RSC was −0.128 MW, and the power flowing through the GSC was the difference between the SCESS power released and itself.

*5.3. Simulation Analysis at a High Wind Speed*

All three groups of the SCESS-DFIGs were set to work at a high wind speed, and the wind speed was 12 m/s. The surge load was added at 2 s, and its values were 10 MW, 15 MW and 20 MW, respectively. The system frequency regulation curve, the P2H and SCESS operation curves, and the GSC and RSC running status curves of the DFIG are shown in Figures 23–26.

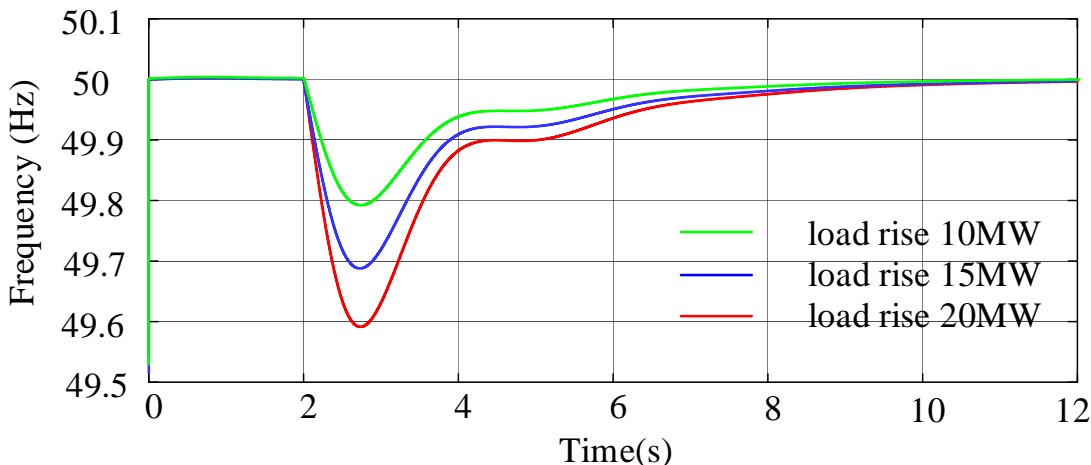

**Figure 23.** System frequency regulation result of high wind speed.

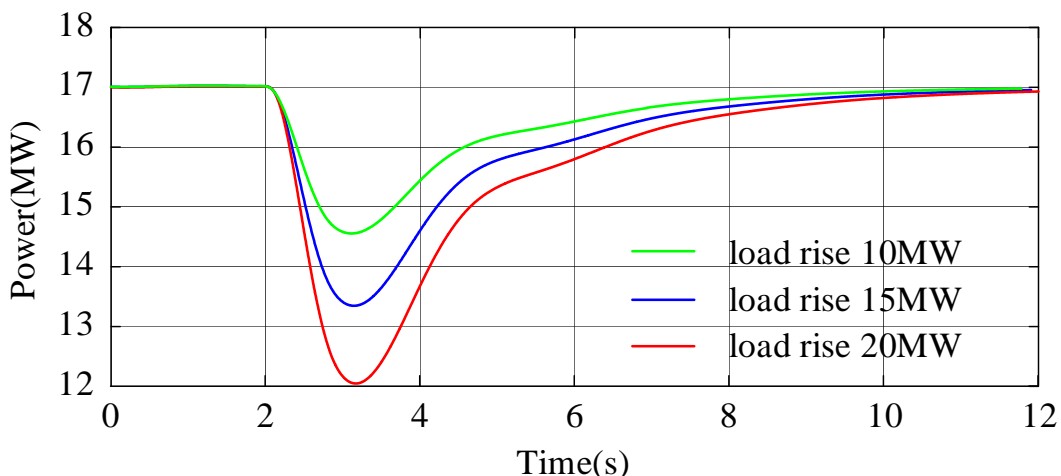

**Figure 24.** Operation curve of P2H at high wind speed.

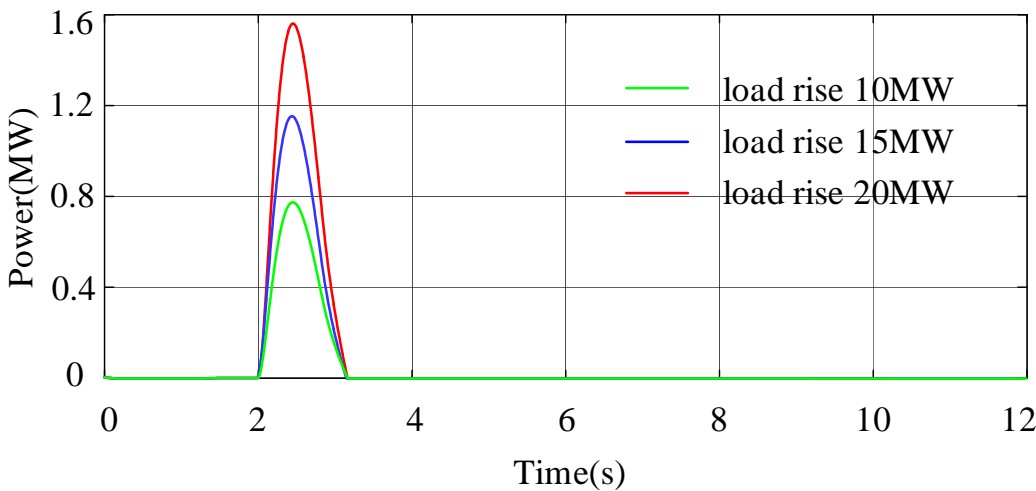

**Figure 25.** Operation curve of SCESS at high wind speed.

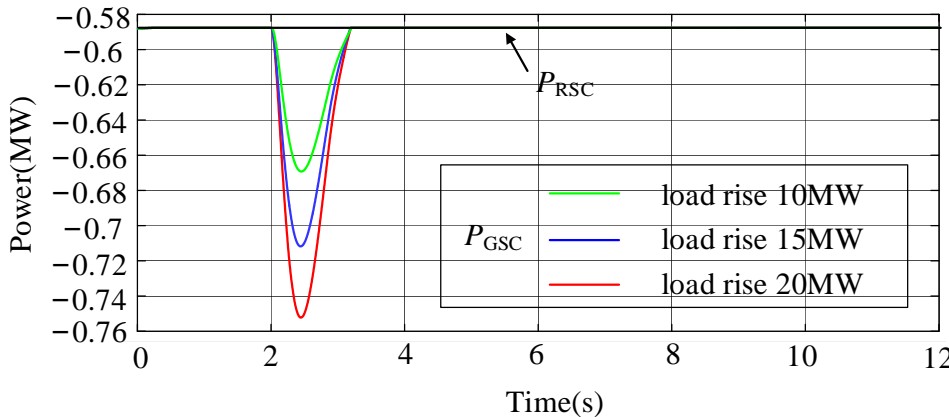

**Figure 26.** GSC and RSC running status curves of DFIG at high wind speed.

As shown in Figures 23–25, the system frequency deviation caused by a load surge at the high wind speed was 0.21 Hz, 0.32 Hz and 0.41 Hz, respectively. When the wind speed was high, the wind turbine maintained the rated power output, a part of the generated power was used for grid-connected power generation, and the rest was used for hydrogen production. At this time, the initial operating power of the P2H was large, and the load shedding ability was strong. At this time, the time constant should, therefore, be reduced and the P2H device should take on more power commands. There was still enough spare capacity of the SCESS after the end of the frequency regulation. As shown in Figure 26, the wind turbine ran in an over-synchronous operation state at this time, the slip rate s was less than 0, the power flowing through the RSC was −0.587 MW, and the power flowing through the GSC was the difference between the SCESS power released and itself.

### 5.4. Simulation Analysis Considering the Aggregation Effect of a SCESS-DFIG-P2H Generating Station

Considering the aggregation effect of the SCESS-DFIG-P2H generating station, three groups of SCESS-DFIGs were set to work at a high wind speed, medium wind speed and low wind speed, respectively. The surge load increased at 2 s, and the value was 15 MW, with the system frequency regulation curve and the frequency regulation-related power curve, when different control strategies were used to participate in the system frequency adjustment, as shown in Figures 27 and 28. Figure 29 depicts the power grid frequency variation curves with the proposed control strategy for the different degrees of surge load (i.e., 10 MW, 15 MW, and 20 MW).

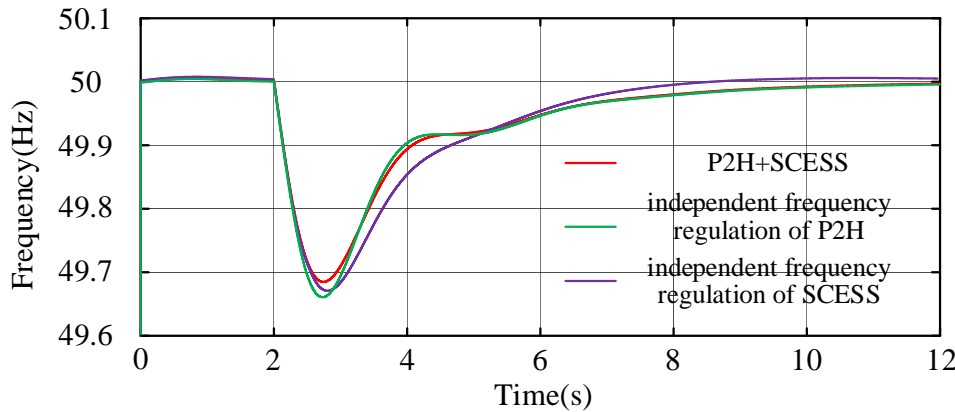

**Figure 27.** System frequency regulation curve under different strategies.

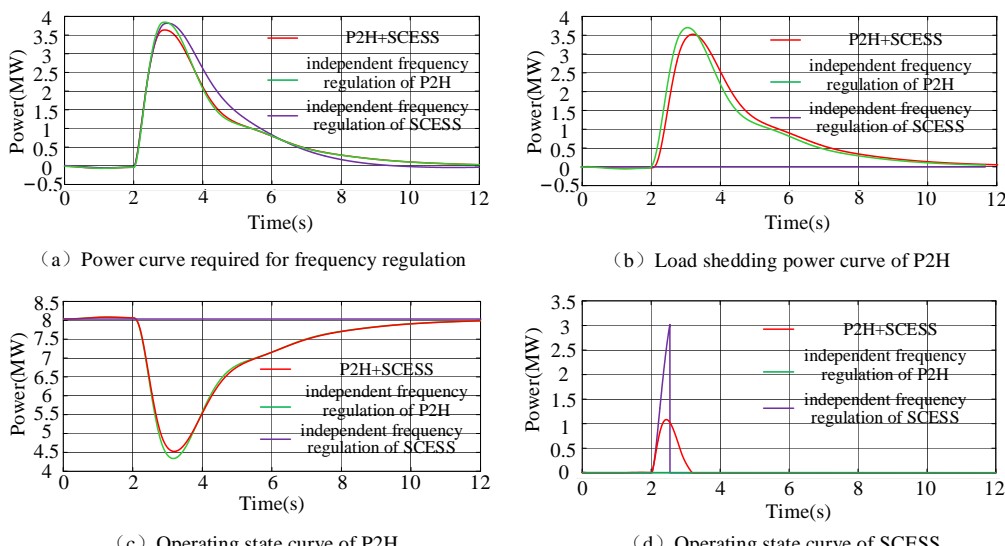

(a) Power curve required for frequency regulation

(b) Load shedding power curve of P2H

(c) Operating state curve of P2H

(d) Operating state curve of SCESS

**Figure 28.** Frequency regulation dependent power curve.

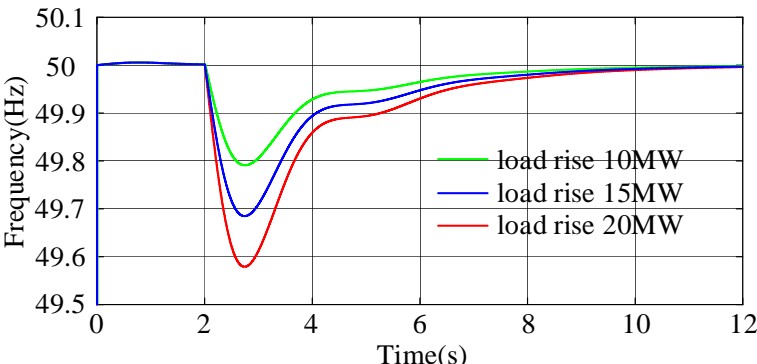

**Figure 29.** System frequency regulation curve under different surge load power.

As shown in Figures 27 and 28, compared with the independent frequency regulation of the P2H device and the SCESS, the coordinated control of the frequency regulation could effectively reduce the system frequency deviation and improve the system's stability. In the independent frequency regulation of P2H, the power commands cannot be tracked in real time due to the P2H's limited working characteristics, while the generated power difference would increase the frequency deviation of the system. In the independent frequency regulation of the SCESS, it could quickly respond to frequency regulation power commands; however, it could not meet the frequency regulation requirements at full time scales due to its capacity limitations. Here, at 2.52 s, the energy of the SCESS was exhausted and it could not participate in the following frequency regulation work. As shown in Figure 29, when the proposed control strategy was used to participate in the system frequency regulation, the maximum value of the system frequency drop did not exceed 0.4 Hz when the load surge range was 10 MW to 20 MW, which is in line with the requirements of the domestic power grid guidelines [7].

Under the different wind speeds and different surge load levels, the system frequency deviation is shown in Table 2. Using the proposed multi-energy coordinated control strategy, the system frequency deviation was less than 0.5 Hz under the different wind speeds, which verifies the effectiveness of the proposed frequency regulation control strategy.

**Table 2.** System frequency deviation.

| Wind Speed/m/s | DFIG Output Power/pu | Exploding Load/MW | min($\Delta f$)/Hz |
|---|---|---|---|
| 8 | 0.312 | 10 | −0.22 |
| | | 15 | −0.32 |
| | | 20 | −0.43 |
| 10 | 0.62 | 10 | −0.21 |
| | | 15 | −0.32 |
| | | 20 | −0.42 |
| 12 | 1 | 10 | −0.21 |
| | | 15 | −0.32 |
| | | 20 | −0.41 |

## 6. Conclusions

In view of the problem that wind turbines in wind storage hydrogen-generating power stations do not have the ability to regulate frequency, and that energy storage devices cannot meet the long-term frequency regulation requirements with independent frequency regulation, in this paper, a method based on the cooperative participation of an electric hydrogen production device and an energy storage device involved in system frequency regulation was proposed, and the following conclusions were drawn from the simulation analysis.

(1) Using the proposed control strategy, at a low wind speed, the system frequency drop values were 0.22, 0.32, and 0.43 Hz when the system suddenly increased the load of 10, 15, and 20 MW, respectively; at a medium wind speed, the system frequency drop values were 0.21, 0.32, and 0.42 Hz when the system suddenly increased the load of 10, 15, and 20 MW, respectively; at a high wind speed, the system frequency drop values were 0.21, 0.32, and 0.41 Hz when the system suddenly increased the load of 10, 15, and 20 MW, respectively. It can be seen that under different wind speeds, the multi-energy coordinated control strategy proposed in this paper should be adopted, and the system frequency deviation would not exceed 0.5 Hz by compensating for the power, so as to maintain the stable operation of a system.

(2) Considering the aggregation effect of the wind storage hydrogen-generating power station, the surge load was added at 2 s, the value was 15 MW, the proposed control strategy was used to participate in the system frequency regulation, and the system frequency deviation was 0.31 Hz; when the electric hydrogen production device was independently frequency regulated, the system frequency deviation was 0.34 Hz; when the super capacitor energy storage device was independently frequency regulated, the system frequency deviation was 0.33 Hz, the super capacitor energy storage device ran out of energy in 2.52 s and it could not participate in the next frequency regulation task. It can be seen that compared with the single energy control strategy, the multi-energy collaborative control strategy reasonably allocated power commands and effectively reduced the degree of the system frequency drop.

To sum up, this paper verifies that the proposed coordinated control strategy can effectively reduce the frequency deviation of a system and improve the stability of that system through a simulation analysis. However, when the wind speed is low, the electric energy will mainly be used for grid-connected power generation, the initial operating power of the electric hydrogen production will be low, and the load shedding capacity will be limited. The control strategy under a low wind speed will be further improved in follow-up work.

**Author Contributions:** Writing—original draft, D.S. and W.Z.; Writing—review & editing, D.S. and J.Y.; Conceptualization, D.S.; Methodology, D.S. and W.Z.; Supervision, D.S.; Data curation, W.Z; Software, W.Z. and J.Y.; Investigation, W.Z; Writing—review & editing, J.Y.; Formal analysis, J.Y.;

Visualization, J.Y.; Resources, J.L.; Funding acquisition, J.L.; Investigation, J.L. All authors have read and agreed to the published version of the manuscript.

**Funding:** This research received no external funding.

**Conflicts of Interest:** The authors declare no conflict of interest.

## Nomenclature

| | |
|---|---|
| $R$ | the radius of the wind turbine blade; |
| $\rho$ | the air density; |
| $v$ | the wind speed; |
| $C_p$ | the wind energy utilization coefficient of the wind turbine; |
| $\lambda$ | the tip speed ratio; |
| $\beta$ | the blade angle; |
| $\Omega$ | the mechanical angular velocity of the DFIG blade; |
| $u_{sd}, u_{sq}, u_{rd}, u_{rq}$ | the stator voltage and rotor voltage components under rotation in the d and q coordinate systems; |
| $i_{sd}, i_{sq}, i_{rd}, i_{rq}$ | the stator current and rotor current components under rotation in the d and q coordinate systems; |
| $\psi_{sd}, \psi_{sq}, \psi_{rd}, \psi_{rq}$ | the stator and rotor magnetic chain components under the rotation of d and q coordinate systems; |
| $\omega_s$ | the electric angular velocity of the d and q coordinate system rotation; |
| $\omega_m$ | the electric angular velocity of the generator input; |
| $L_\sigma$ | the leakage inductance between the stator and rotor under the d and q coordinate system rotation; |
| $L_m$ | the mutual inductance between the stator and rotor under the d and q coordinate system rotation; |
| $L_R$ | the sum of the leakage and mutual inductance between the stator and rotor under the d and q coordinate system rotation; |
| $E_{ks}$ | the rotor kinetic energy of the generator at the rated speed; |
| $J$ | the moment of inertia of the generator; |
| $\omega_S$ | the rated speed of the generator; |
| $S_N$ | the rated capacity of the generator; |
| $\sum_{i=1}^{n} E_{w.i} = 0$ | the sum of the blade kinetic energy of n wind turbines in the SCESS-DFIG-P2H generating station system; |
| $E_S$ | the energy that can be handled by the energy storage device; |
| $E_P$ | the energy provided by the electric hydrogen production device as a controllable load; |
| $S_{NW}$ | the wind farm rated capacity; |
| $\Delta P_e^*$ | the per-unit value of generator electromagnetic change power; |
| $\omega^*$ | the per-unit value of rotor speed; |
| $\Delta T_e^*$ | the per-unit value of generator electromagnetic torque change; |
| $f^*$ | the system frequency per-unit value; |
| $f^*(t)$ | the per-unit value of the frequency of the system at time $t$; |
| $f^*(t + \Delta t)$ | the per-unit value of the frequency at time $t + \Delta t$; |
| $\Delta E^*$ | the per-unit value of the energy that can be released by the energy storage device and P2H system in time; |
| $P_N$ | the rated power of the generator; |
| $K$ | the droop coefficient of the generator; |
| $U_{cell}$ | the single cell voltage of the electrolytic cell; |
| $I_{el}$ | the current flowing through the electrolytic cell; |
| $U_{rev}$ | the reversible cell voltage; |
| $r_1, r_2$ | the ohmic parameters of the electrolyte; |
| $s_1, s_2, s_3, t_1, t_2, t_3$ | the electrodes overvoltage parameter; |

| | |
|---|---|
| $A_{el}$ | the electrode area; |
| $T_{el}$ | the electrolyte temperature; |
| $N_{el}$ | the number of batteries connected in series in the electrolytic cell; |
| $U_{el}$ | the voltage of the electrolytic cell; |
| $\eta_F$ | the faradaic efficiency; |
| $V_{H_2}$ | the hydrogen production rate of the electrolysis cells; |
| $P_{el,t}$ | the power command that the electrolytic cell needs to undertake at time $t$; |
| $\Delta P^{max}$ | the power change threshold per unit time; |
| $D_{el}$ | the control signal of the P2H device; |
| $K_{p1}$ and $K_{p2}$ | the proportional adjustment gain and integral adjustment gain of the current loop, respectively; |
| $I_{elref}$ and $I_{el}$ | the reference value and feedback value of the electrolysis cell current; |
| $U_{elc}$ | the voltage value of the DC bus of the P2H system; |
| $\tau$ | the time constant; |
| $D$ | the step amplitude; |
| $c(0)$ and $r(0)$ | represent the initial value of the state in the system and the initial value of the input; |
| $D_{scbuck}$ | the control signal of the bidirectional DC–DC converter of the SCESS in Buck mode; |
| $K_{s1}$ and $K_{s2}$ | the proportional adjustment gain and integral adjustment gain of the current loop, respectively; |
| $I_{scref}$ and $I_{sc}$ | the reference and feedback values of the current flowing through the SCESS; |
| $U_{sc}$ | the voltage value across the SCESS; |
| $U_{dc}$ | the DC bus voltage value of the DFIG; |
| $D_{scboost}$ | the control signal when the bidirectional DC–DC converter of the SCESS is in Boost mode. |

## Appendix A

As shown in Figure A1, the prime mover generates the mechanical torque $T_M$, and the generator changes the output power of the synchronous unit through the electromagnetic torque $T_E$. There is a coupling relationship between the rotor angular velocity of the synchronous unit $\omega_r$ and the grid frequency $f$. The drop of the grid frequency will cause the decrease of the rotor angular velocity. The rotor motion equation is shown in Equation (A1):

$$J\frac{d\omega_r}{dt} = T_M - T_E \tag{A1}$$

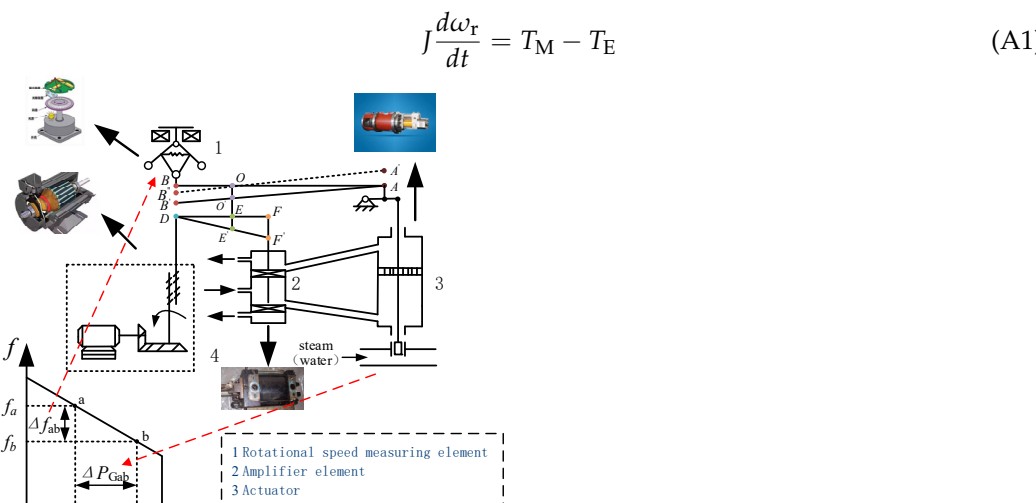

**Figure A1.** Droop mechanism analysis of a synchronous set.

Among them, $T_M$ is the mechanical torque of the prime mover; $T_E$ is the electromagnetic torque of the generator. The mechanical torque of the prime mover remains unchanged, and a reduction in the rotor electrical angular velocity $\omega_r$ will lead to an increase

in $T_E$, resulting in an electromagnetic torque increment $\Delta T_{E_1}$, which in turn damps the rapid change of the system frequency, that is, the inertia response of the synchronous unit.

As shown in Figure A1, the frequency decreases from point a to point b, and the rotor speed of the generator set decreases, so that the centrifugal force of the centrifugal pendulum decreases, and the sleeve moves down from point B to point B'. At this time, the servomotor does not move, and the lever AOB rotates around point A, so that point O drops to point O', then point F slides down, and the piston of the mechanism 2 pilot valve moves down, opening the oil hole leading to the servomotor of mechanism 3. The pressure oil enters the servomotor and pushes the piston upward, which increases the opening of the regulating valve, increases the intake throttle flow, and increases the input power of the prime mover. At this time, the prime mover in the synchronous unit generates a mechanical torque increment $\Delta T_{M_1}$, which causes the electromagnetic torque of the motor to increase $\Delta T_{E_2}$, and the speed of the unit will begin to rise.

As the speed rises, the sleeve starts to move up from point B', and point A also moves up. When the O point and DEF return to the initial position, the pilot valve re-blocks the two oil holes, and the servomotor piston also returns to stability. The adjustment process ends at this time.

## Appendix B

**Table A1.** Simulation parameters of wind turbines and a synchronous unit.

| | Project | Value |
|---|---|---|
| | Number of blades | 3 |
| | Blade diameter | 90 m |
| | Working wind speed | 3~25 m/s |
| | Blade quality | 55,000 kg |
| | Moment of inertia | 55,687,500 kg·m$^2$ |
| Wind turbines | Gear ratio | 1:77.44 |
| | Gearbox mass | 18,500 kg |
| | RSC rated power | 0.25 pu |
| | GSC rated power | 0.25 pu |
| | DC converter rated power | 0.25 pu |
| | Number of pole pairs | 3 |
| | Rated speed | 1000 r/min |
| | Generator power | 0.3 pu |
| | Differential regulation coefficient of generator | 11 |
| Synchronous generator | Generator power frequency static characteristic coefficient | 20 |
| | Generator inertia time constant | 5 |
| | Generator rated power | 100 MW |

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
