# Peer review of "Research on the Primary Frequency Regulation Control Strategy of a Wind Storage Hydrogen-Generating Power Station"

_electronics, doi:10.3390/electronics11223669_

Round 1

Reviewer 1 Report

1. In section 2.1 authors must refer some more details for the DFIG system, which they assume in their paper (eg. dynamic equations or they could describe the operational situation of it giving some references).

2. Authors should define acronyms before they use them for the first time.

3. Line 472: Figures 17-19?

4. In section 5.1: Why have authors chosen 8m/s for wind speed and not lower? Simulation scenarios assuming more extreme wind velocity values must be conducted (both for low speed and high speed scenario).

5. Disadvantages of the proposed method should be referred.

6. Maybe some more simulation results (presenting additional operational variables waveforms) should be added.

Reviewer 2 Report

In this manuscript, a multi-energy coordinated frequency regulation control strategy is introduced. There some significant vague points that should be addressed by the authors.

1. The phrase “SCESS-DFIG-P2H” must be introduced when it is first used in the text.

2. The abstract of the manuscript should be modified. The current version of abstract cant introduce the main novelty and purpose of this manuscript.

3. The authors claim that a single energy storage device cannot meet the full time scale frequency regulation requirements. Why?

4. At table 2, there are two rows for “Transmission quality”. The value for first Transmission quality is indicated 18500 Kg and the value for the second Transmission quality is indicated 0.25pu. What is the “Transmission quality”?

5. There is not any result from the operation of DFIG. Some results such as DC link voltage of the DFIG, current of RSC as well as GSC and the voltage of PCC should be introduced to verify the proper operation of DFIG system.

Reviewer 3 Report

This reviewer suggests the following points to improve the paper quality:

1.       Please try to avoid the acronyms in Title and Keywords. Also limit acronyms in Abstract and Conclusion sections. The acronyms/abbreviated terms should be defined/expanded at their first appearance.

2.       Try to include the nomenclature of all symbols used in the work, before Introduction section for better readability. Also include the design values of system parameters considered in this work after Conclusion section as Appendix.

3.       Try to redraft the Introduction section, with background, challenges, literature review, scopes, motivation, contributions, and organization of paper. Highlight the novelties/major contribution of the work prior to organization pf paper in brief (preferably in 3-bulleted points), Expand the literature review part, including some recent (of last 3-4 years) papers in the similar field, such as, doi: 10.1109/ACCESS.2020.3025292 , 10.1016/j.seta.2021.101622 , 10.1109/ACCESS.2019.2946192, 10.1049/joe.2018.8504, and so on.

4.       Try to emphasize more on the problem statement and validate the system model with comparative analysis.

5.       Try to quote all equations at appropriate texts with suitable citations (if adopted from published work).

6.       Why the Author considered SCESS-P2H units with DFIG wind generating station? Justify your answer by comparative analysis with classical/contemporary methods.

7.       The validation of the proposed method should be provided with a specific case study to support the claim.

8.       Results should be supported with more discussion for different scenarios of wind variations.

9.       Redraft the Conclusion with numerical evidence to support your claim. Also include at least one future scope to it.

10.   Proofread the entire manuscript to rectify some existing typos and grammatical errors.

Round 2

Reviewer 1 Report

I think that the paper can be published now.

Reviewer 2 Report

The manuscript has been sufficiently improved to warrant publication in Electronics from my point of view.